# SAMHD1 is a key regulator of the lineage-specific response of acute lymphoblastic leukaemias to nelarabine

Tamara Rothenburger[1], Katie-May McLaughlin[2], Tobias Herold [3,4], Constanze Schneider[1,10], Thomas Oellerich[5,6,7], Florian Rothweiler[1], Andrew Feber [8], Tim R. Fenton [2], Mark N. Wass [2], Oliver T. Keppler[9], Martin Michaelis [2✉] & Jindrich Cinatl Jr. [1✉]

The nucleoside analogue nelarabine, the prodrug of arabinosylguanine (AraG), is effective against T-cell acute lymphoblastic leukaemia (T-ALL) but not against B-cell ALL (B-ALL). The underlying mechanisms have remained elusive. Here, data from pharmacogenomics studies and a panel of ALL cell lines reveal an inverse correlation between nelarabine sensitivity and the expression of *SAMHD1*, which can hydrolyse and inactivate triphosphorylated nucleoside analogues. Lower SAMHD1 abundance is detected in T-ALL than in B-ALL in cell lines and patient-derived leukaemic blasts. Mechanistically, T-ALL cells display increased *SAMHD1* promoter methylation without increased global DNA methylation. SAMHD1 depletion sensitises B-ALL cells to AraG, while ectopic SAMHD1 expression in SAMHD1-null T-ALL cells induces AraG resistance. SAMHD1 has a larger impact on nelarabine/AraG than on cytarabine in ALL cells. Opposite effects are observed in acute myeloid leukaemia cells, indicating entity-specific differences. In conclusion, *SAMHD1* promoter methylation and, in turn, *SAMHD1* expression levels determine ALL cell response to nelarabine.

[1] Institut für Medizinische Virologie, Klinikum der Goethe-Universität, Paul Ehrlich-Straße 40, 60596 Frankfurt am Main, Germany. [2] School of Biosciences, University of Kent, Canterbury CT2 7NJ, UK. [3] Department of Medicine III, University Hospital, LMU Munich, Marchioninistraße 15, 81377 Munich, Germany. [4] Research Unit Apoptosis in Hematopoietic Stem Cells, Helmholtz Zentrum München, German Research Center for Environmental Health (HMGU), Feodor-Lynenstraße 21, 81377 Munich, Germany. [5] Department of Medicine II, Hematology/Oncology, Goethe-Universität, Theodor-Stern-Kai 7, 60590 Frankfurt am Main, Germany. [6] Frankfurt Cancer Institute, Goethe University, Theodor-Stern-Kai 7, 60590 Frankfurt am Main, Germany. [7] German Cancer Consortium/German Cancer Research Center, Im Neuenheimer Feld 280, 69120 Heidelberg, Germany. [8] Division of Surgery and Interventional Science, University College London, Gower Street, London WC1E 6BT, UK. [9] Faculty of Medicine, Max von Pettenkofer Institute, Virology, LMU München, Pettenkoferstraße 9a, 80336 Munich, Germany. [10] Present address: Department of Medicine II, Hematology/Oncology, Goethe-Universität, Frankfurt am Main, Germany; Frankfurt Cancer Institute, Goethe University, Theodor-Stern-Kai 7, 60590 Frankfurt am Main, Germany. ✉email: M.Michaelis@kent.ac.uk; Cinatl@em.uni-frankfurt.de

Acute lymphoblastic leukaemia (ALL) cells originate from precursor lymphoid T- (T-ALL) and B-cells (B-ALL). In children, ALL is the most common cancer associated with high cure rates of about 85%. In adults, ALL accounts for 15–25% of acute leukaemias and is associated with a less favourable outcome[1–4]. Among ALLs, T-ALL is responsible for ~15% of paediatric ALLs and 25% of adult ALLs[3]. Nelarabine displays selective activity in T-ALL over B-ALL and is used for the treatment of relapsed and refractory T-ALL but not routinely for the treatment of B-ALL[3,5–13]. However, the molecular mechanisms underlying this difference remain elusive. Moreover, nelarabine therapy can be associated with irreversible life-threatening neurotoxicity[8,14]. Hence, biomarkers indicating patients who are most likely to benefit from nelarabine therapy are needed.

Here, we used an approach that combined data from the large pharmacogenomics screens Cancer Therapeutics Response Portal (CTRP)[15], Cancer Cell Line Encyclopedia (CCLE)[16], and Genomics of Drug Sensitivity in Cancer (GDSC)[17] with data from an ALL cell line panel derived from the Resistant Cancer Cell Line (RCCL) collection and patient data to investigate the mechanisms underlying the discrepancy in the nelarabine sensitivity between T-ALL and B-ALL.

The results show that low expression of Sterile alpha motif and histidine-aspartic acid domain-containing protein 1 (SAMHD1) in T-ALL cells is a key determinant of nelarabine sensitivity and that SAMHD1 is a potential biomarker and therapeutic target for the improvement of nelarabine-based therapies for both T-ALL and B-ALL patients.

## Results

**Gene expression comparison between T-ALL and B-ALL cells.** To identify potential differences between T-ALL and B-ALL that may explain the observed discrepancies in nelarabine sensitivity, we started by analysing data derived from the large pharmacogenomics databases CCLE[16], Cancer Therapeutics Response Portal (CTRP)[15], and Genomics of Drug Sensitivity in Cancer (GDSC)[17]. The CCLE contained data derived from 34 leukaemia cell lines (18 B-ALL, 16 T-ALL) and the GDSC from 38 leukaemia cell lines (21 B-ALL, 17 T-ALL), with an overlap of 19 cell lines (Supplementary Table 1). The CCLE and CTRP used the same cell line panel for their studies[15].

Nelarabine was tested in 24 ALL (11 B-ALL, 13-T-ALL) cell lines in the CTRP (Supplementary Table 2). In agreement with the available literature[5,6,18,19], nelarabine displayed higher activity in T-ALL than in B-ALL cell lines (Supplementary Fig. 1A, Supplementary Table 2).

Initially, we compared transcriptomics data (mRNA abundance) between T-ALL and B-ALL cell lines. Substantial proportions of transcripts displayed significant differences ($P < 0.05$) in their abundance levels between T-ALL and B-ALL cells in the GDSC (3,998/ 22.5% of 17,735 transcripts), CCLE (8,498/ 42.1% of 20,172 transcripts), and CTRP (4,507/ 24.3% of 18,539 transcripts) (Supplementary Data 1). Gene expression heatmaps illustrated these differences (Supplementary Fig. 1), but manual analysis of the top differentially regulated genes did not result in the identification of candidate genes, whose expression seemed likely to be responsible for the observed differences in nelarabine sensitivity.

A pathway analysis using the PANTHER database (Protein ANalysis THrough Evolutionary Relationships, http://pantherdb.org)[20] also did not reveal processes that may underlie the increased nelarabine sensitivity of T-ALL cells (Supplementary Fig. 1, Supplementary Data 2). As expected, B-cell- and T-cell-specific processes featured prominently among the most strongly differentially regulated pathways.

**SAMHD1 levels correlate with nelarabine resistance.** The correlation of transcriptomics data with the nelarabine drug response, represented as AUC, identified SAMHD1 as the gene, whose expression displayed the most significant direct correlation (Supplementary Data 3). Analysis of SAMHD1 expression exclusively in either the B-ALL or T-ALL subset also showed a highly significant direct correlation with the nelarabine AUC (Supplementary Data 3). Furthermore, when we correlated drug AUCs with SAMHD1 expression, nelarabine displayed the most significant direct correlation with SAMHD1 expression across all ALL cell lines, the second most significant direct correlation with SAMHD1 expression in the B-ALL cell lines, and the third most significant direct correlation with SAMHD1 expression in the T-ALL cell lines (Supplementary Data 4).

**SAMHD1 levels are lower in T-ALL than in B-ALL cells.** SAMHD1 is a deoxynucleotide triphosphate (dNTP) hydrolase that cleaves physiological dNTPs and triphosphorylated nucleoside analogues[21–25]. It was previously shown to interfere with the activity of anti-cancer nucleoside analogues including nelarabine[23,24,26]. If SAMHD1 was responsible for the differences observed in nelarabine sensitivity between T-ALL and B-ALL, T-ALL cells would be expected to express lower levels of SAMHD1. Indeed, the SAMHD1 expression (mRNA abundance) levels were significantly lower in T-ALL than in B-ALL cell lines in all three databases (Fig. 1a). Similar findings were detected in a gene expression dataset derived from blasts of 306 ALL (222 B-ALL, 84 T-ALL) patients[27,28] (Fig. 1b). Further analysis revealed a reduced expression of SAMHD1 in T-ALL in general but more pronounced in the thymic and mature immunophenotypic subtype (Supplementary Fig. 2A). On the genetic level, some B-ALL subgroups like for example Philadelphia (Ph)-like patients display a gene expression pattern of SAMHD1 that is equally low as seen in T-ALL (Supplementary Fig. 2B).

**SAMHD1 is a determinant of nelarabine sensitivity in ALL.** A number of other gene products have been described to be involved in the transport, activation, and metabolism of nucleoside analogues such as nelarabine, including DCK, DGUOK, SLC29A1 (ENT1), SLC29A2 (ENT2), NT5C, NT5C2, PNP, RRM1, RRM2 and SLC22A4 (OCTN1)[19,29]. While statistically significant differences in the expression of some of the respective genes were noted between B-ALL and T-ALL cell lines in some of the three datasets, none was consistent across all three and none was as robust as in the expression of SAMHD1 (Fig. 1c, Supplementary Fig. 3). In patient samples, SAMHD1 also displayed the most significant difference in expression levels between B-ALL and T-ALL (Supplementary Fig. 3). Moreover, only the expression of SAMHD1 correlated with the nelarabine AUC in the CTRP dataset (Fig. 2, Supplementary Fig. 4). This shows that SAMHD1 is a critical determinant of nelarabine efficacy in ALL and that low SAMHD1 levels critically contribute to the specific nelarabine sensitivity of T-ALL cells.

**SAMHD1 is no determinant of cytarabine sensitivity in ALL.** Cellular SAMHD1 levels have previously been shown to critically determine cytarabine efficacy in acute myeloid leukaemia (AML) cells[23,24,30] and SAMHD1 expression levels are lower in T-ALL than in AML cells (Supplementary Fig. 5). The CTRP and GDSC contained data on cytarabine activity. In contrast to AML cells, however, there was no difference in the cytarabine sensitivity between B-ALL and T-ALL cell lines and no correlation between SAMHD1 expression and cytarabine sensitivity in ALL cells (Fig. 2, Supplementary Fig. 6). Hence, the effect of SAMHD1 on nucleoside analogue activity depends on the tissue context.

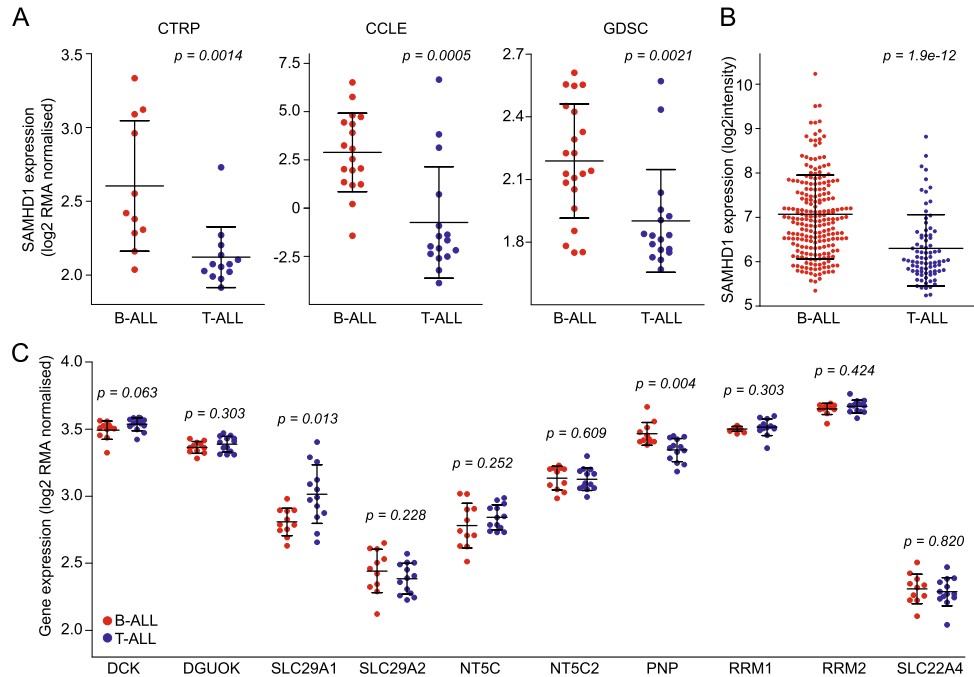

**Fig. 1 SAMHD1 levels differ between T-ALL and B-ALL.** Comparison of SAMHD1 expression (mRNA abundance) levels in T-ALL and B-ALL cell lines from the CTRP, CCLE, and GDSC (**a**) and in blasts from leukaemia patients (**b**). **c** Comparison of the expression of other genes known to affect nucleoside analogue activity based on CTRP data. Respective CCLE and GDSC data are provided in Supplementary Fig. 2. *p-values for the comparison B-ALL vs. T-ALL.

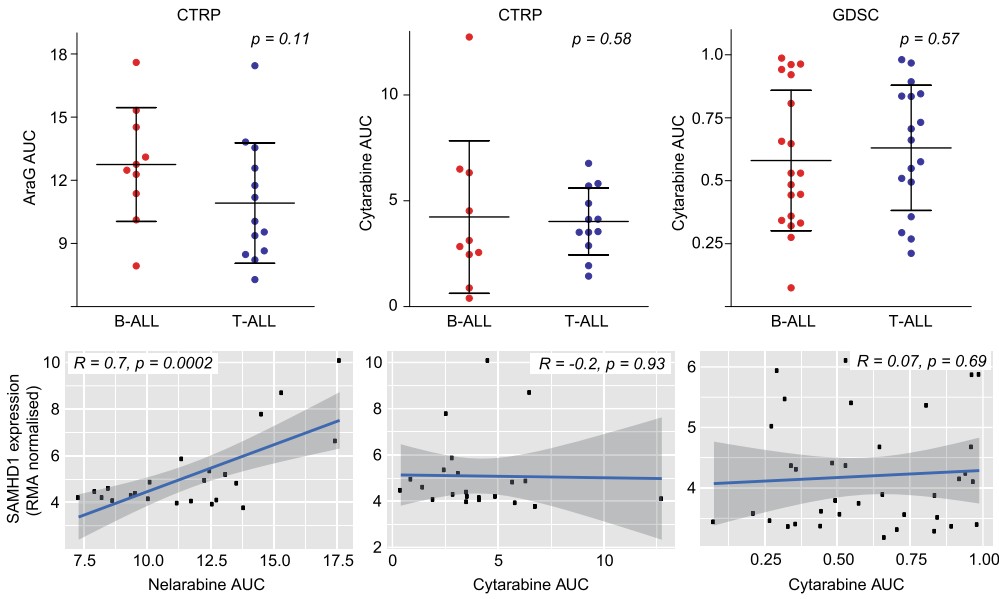

**Fig. 2 Comparison of nelarabine (CTRP) and cytarabine (CTRP, GDSC) sensitivity between B-ALL and T-ALL cell lines and correlation of SAMHD1 mRNA levels with the nelarabine and cytarabine sensitivity (expressed as AUC) across all B-ALL and T-ALL cell lines.** Pearson's r values and respective p-values are provided. Respective data on the correlation of *SAMHD1* expression with drug sensitivity exclusively for B-ALL and T-ALL cell lines are provided in Supplementary Fig. 3 (nelarabine) and Supplementary Fig. 4 (cytarabine).

**SAMHD1 mRNA levels reflect protein levels in ALL cell lines**. To further investigate the role of SAMHD1 on nelarabine and cytarabine efficacy in ALL, we assembled a panel consisting of 15 B-ALL and 11 T-ALL cell lines from the RCCL collection[31] (Supplementary Table 3). Firstly, we investigated the extent to which cellular SAMHD1 mRNA levels are indicative of cellular protein levels. Western blot analyses confirmed that the RCCL

T-ALL cell lines generally display lower SAMHD1 protein levels than the RCCL B-ALL cell lines (Fig. 3a, Supplementary Fig. 7). However, quantitative western blot analysis and quantitative PCR (qPCR) showed that cellular SAMHD1 mRNA levels do not always directly correlate with cellular SAMHD1 protein levels (Fig. 3b). This is likely to reflect the complexity of the regulation of protein levels, which are determined by transcription and

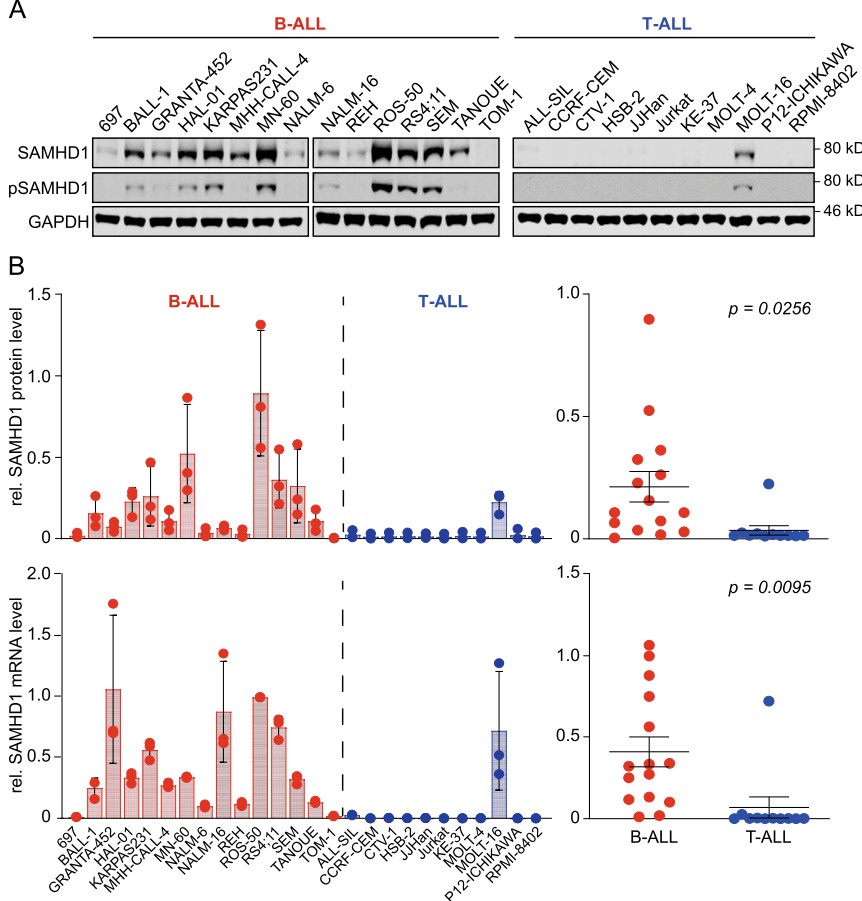

**Fig. 3 SAMHD1 protein and mRNA levels in the RCCL panel of B-ALL and T-ALL cell lines. a** Representative Western blots indicating protein levels of total SAMHD1 and phosphorylated SAMHD1 (p-SAMHD1). GAPDH was used as loading control. **b** Quantitative SAMHD1 protein levels are shown as means ± SD from three independent experiments (quantified using near-infrared Western blot images to determine the ratio SAMHD1/ GAPDH relative to the positive control THP-1, an acute myeloid leukaemia cell line characterised by high cellular SAMHD1 levels [Schneider et al.[30]]. SAMHD1 mRNA abundance levels are shown as means ± SD from three technical replicates (as determined by qPCR, relative to cell line ROS-50) in B-ALL and T-ALL cell lines. Unpaired two-tailed Student's t-tests were used to compare means (represented as horizontal lines ± SEM) of SAMHD1 protein or mRNA levels in B-ALL and T-ALL cells.

translation efficacy, factors that control mRNA stability (e.g. microRNAs and proteins that control mRNA degradation), and post-translational modifications that promote (proteasomal) protein degradation[32–37]. Moreover, mutations may affect SAMHD1 function, as demonstrated in patients with chronic lymphocytic leukaemia and colorectal cancer[38,39]. However, the only ALL cell line with a *SAMHD1* mutation was Jurkat, which harboured an R611* nonsense mutation based on GDSC data. SAMHD1 mRNA and protein levels in the RCCL are correlated with SAMHD1 mRNA levels in the corresponding cell lines from CTRP, CCLE and GDSC (Supplementary Fig. 8). Hence, SAMHD1 mRNA levels, largely predict SAMHD1 protein levels, which is in line with previous findings[30].

Next, the sensitivity of the RCCL ALL cell lines was tested against arabinosylguanine (AraG), the product of the prodrug nelarabine[40] and cytarabine. The results were in agreement with the CTRP data showing that T-ALL cell lines were significantly more sensitive to AraG than B-ALL cell lines (Fig. 4, Supplementary Table 3). Notably, there was a significant correlation between the nelarabine AUCs in the CTRP and the AraG IC50s in the RCCL panel among the cell lines that were present in both datasets (Supplementary Fig. 9). In contrast to the CTRP and GDSC data that had not indicated a difference between the

cytarabine sensitivity of T-ALL- and B-ALL- cells, T-ALL cell lines displayed a trend indicating increased sensitivity to cytarabine ($P = 0.055$) (Fig. 4). SAMHD1 protein levels displayed a significant correlation with the AraG concentrations that reduced cell viability by 50% (IC50) in all ALL cell lines and the lineage-specific subanalyses (Fig. 4). In contrast, a significant correlation between SAMHD1 protein levels and cytarabine activity was only detected across all ALL cell lines but not when only B-ALL or T-ALL cell lines were considered (Fig. 4). SAMHD1 mRNA levels were correlated with the AraG IC50 across all ALL cell lines and T-ALL cell lines but not B-ALL cell lines ($P = 0.1335$) (Fig. 4). No significant correlation was detected between the SAMHD1 mRNA levels and the cytarabine IC50 in the RCCL ALL cell lines (Fig. 4).

Taken together, these results confirm the CTRP data in showing that cellular SAMHD1 levels determine ALL sensitivity against AraG, the product of nelarabine, and that low SAMHD1 levels in T-ALL cells are associated with specific nelarabine/ Ara-G activity in this lineage. In contrast to the CTRP and GDSC data, the additional experimental analyses in the RCCL ALL cell line panel suggest that SAMHD1 levels may also affect cytarabine activity in ALL, albeit to a lower degree than AraG activity.

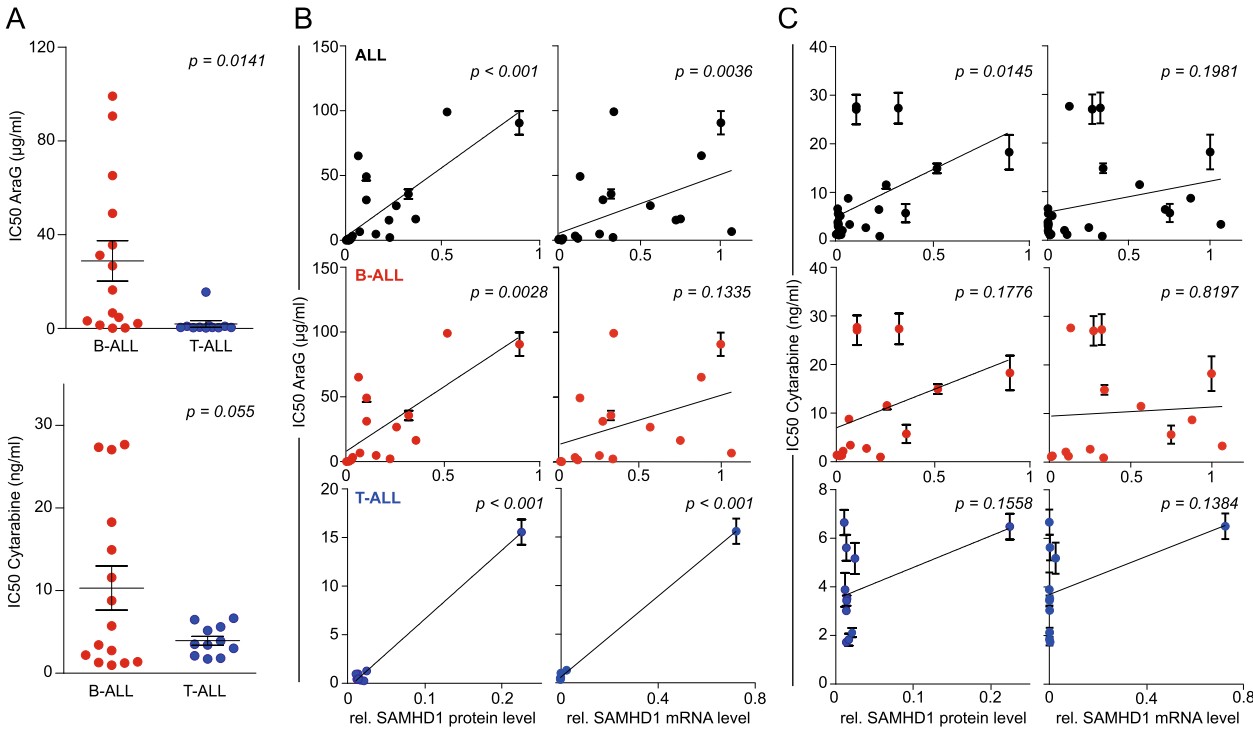

**Fig. 4 AraG and cytarabine concentrations that reduce the viability of the RCCL ALL cell lines by 50% (IC50) and correlation of the IC50s with the cellular SAMHD1 protein or mRNA levels.** Numerical data are provided in Supplementary Data 4. Closed circles and error bars represent means ± SD of three independent experiments, each performed in three technical replicates. Linear regression analyses were performed using GraphPad Prism.

**SAMHD1 depletion sensitises ALL cells to AraG.** To further investigate the functional role of SAMHD1 in determining AraG and cytarabine activity in ALL cells, we depleted SAMHD1 using virus-like particles containing Vpx as previously described[30,41]. Vpx is a protein encoded by HIV-2 and certain SIV strains that mediates proteasomal SAMHD1 degradation[41–43]. Vpx virus-like particles resulted in the sensitisation of ALL cells to AraG and cytarabine but exerted much more pronounced effects on the activity of AraG (Fig. 5a).

In the ALL cell lines MHH-CALL-4, SEM, and TANOUE, the AraG IC50s were between 37.5- and 101-fold lower following exposure to Vpx virus-like particles compared to Vpr virus-like particles, which served as negative controls. In contrast, Vpx virus-like particles only reduced the cytarabine IC50s by 5- and 7-fold lower in these cell lines.

**Different role of SAMHD1 as resistance factor in ALL and AML.** In AML cells, SAMHD1 has been described as a critical regulator of cytarabine activity[30]. Since Vpx virus-like particle-mediated SAMHD1 depletion had resulted in a more pronounced sensitisation of ALL cells to AraG than to cytarabine, we further compared the effect of the presence or absence of functional SAMHD1 on the activity of these structurally related nucleoside analogues in these two types of acute leukaemia. Cell models included the SAMHD1-expressing AML cell line THP-1 and its subline, in which the SAMHD1 gene had been disrupted by CRISPR/Cas9 (THP-1- KO). Further, we investigated the SAMHD1 low/ non- expressing cell lines HEL (AML) and Jurkat (T-ALL) and their respective sublines transduced either with wild-type (WT) SAMHD1 or the triphosphohydrolase-defective mutant SAMHD1-D311A. In the AML cell lines, absence of functional SAMHD1 was associated with a 60-fold (THP-1/ THP-1-KO) and 6583-fold (HEL-SAMHD1_WT/HEL-SAMHD1_D311A) sensitisation to cytarabine, but only a 5.6- and 6.0-fold sensitisation to

AraG (Fig. 5b). The T-ALL cell line Jurkat-SAMHD1_D311A was 101 times more sensitive to AraG than Jurkat-SAMHD1_WT, while JURKAT-SAMHD1_D311A was only 10 times more sensitive to cytarabine (Fig. 5b). In summary, SAMHD1 activity critically regulates cytarabine activity but has a much lower impact on AraG in AML cells. The opposite effect is observed in ALL cells, in which SAMHD1 crucially determines AraG activity but exerts substantially less pronounced effects on cytarabine activity. This further confirms that the cellular background critically determines the importance of SAMHD1 as regulator of nucleoside activity.

**High SAMHD1 promoter methylation in T-ALL cell lines.** SAMHD1 levels may be regulated by SAMHD1 promoter methylation in leukaemia cells[44,45]. Therefore, we compared SAMHD1 promoter methylation in T-ALL and B-ALL cell lines through amplification of a single PCR product (993-bp) corresponding to the promoter sequence after HpaII digestion. Results indicated that the SAMHD1 promoter was methylated in all T-ALL cell lines but one (MOLT-16) (Fig. 6a), which was the only T-ALL cell line characterised by high SAMHD1 mRNA and protein levels (Fig. 3) and low AraG sensitivity (Supplementary Table 3). In contrast, SAMHD1 promoter methylation was only observed in two out of 15 B-ALL cell lines (NALM-6, TOM-1) (Fig. 6a). In agreement, SAMHD1 promoter methylation was also significantly higher in T-ALL than in B-ALL cells in the GDSC and inversely correlated with SAMHD1 expression (Fig. 6b). Notably, global DNA methylation did not differ between T-ALL and B-ALL cell lines (Fig. 6c), suggesting lineage-specific differences. Taken together, this suggests that the differences in cellular SAMHD1 levels observed between T-ALL and B-ALL cell lines are to a large extent caused by differences in SAMHD1 promoter methylation.

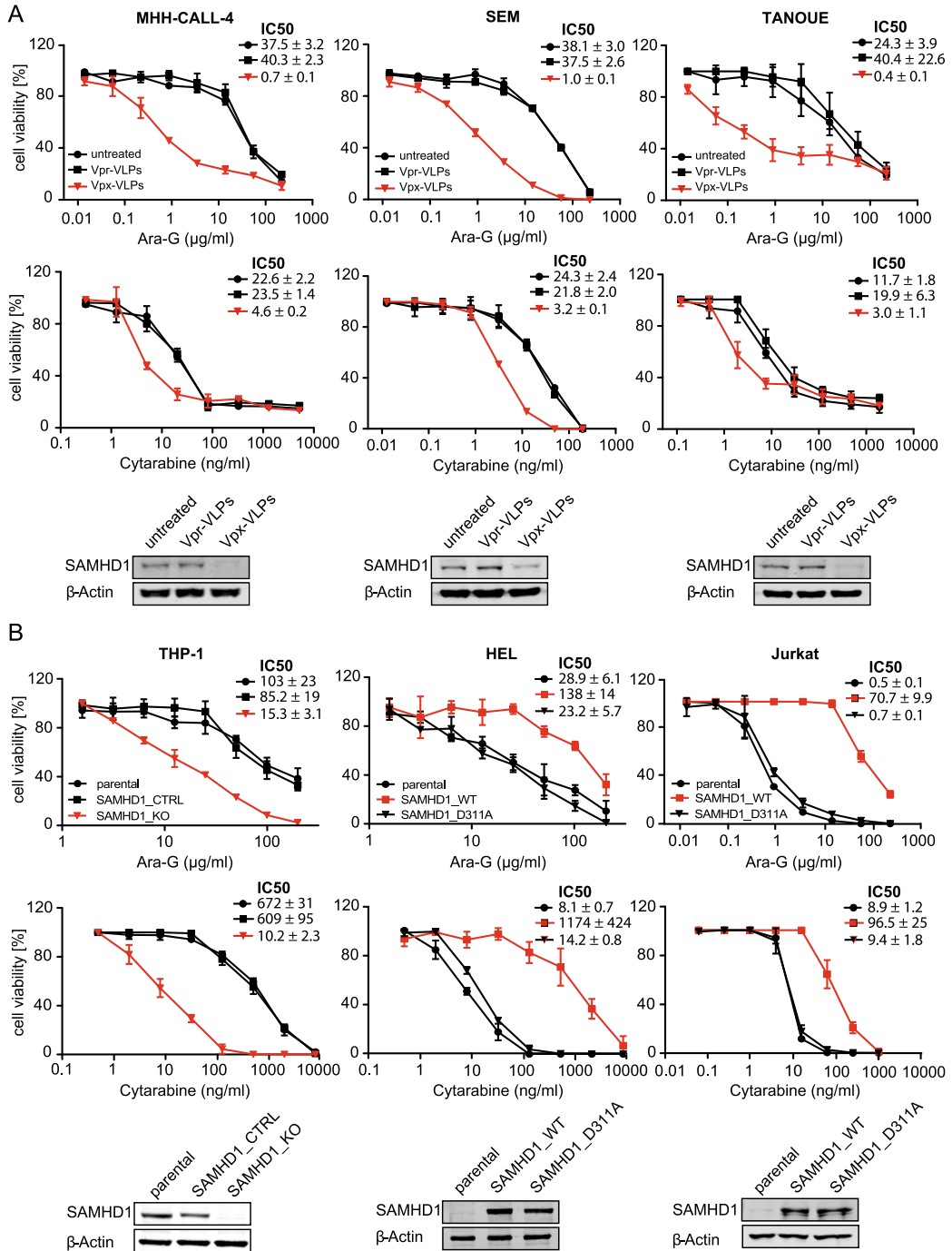

**Fig. 5 Effect of SAMHD1 on nelarabine and cytarabine sensitivity in ALL and AML cells. a** Dose-response curves of AraG- and cytarabine-treated ALL cell lines in the absence or presence of Vpx virus-like particles (cause SAMHD1 depletion), or Vpr virus-like particles (negative) controls. Concentrations that reduce ALL cell viability by 50% (IC50s) and Western blots confirming SAMHD1 depletion are provided. Each symbol represents the mean ± SD of three technical replicates of one representative experiment out of three. **b** Effects of AraG and cytarabine on AML cell lines in the absence or presence of functional SAMHD1. In the *SAMHD1*-expressing AML cell line THP-1, the *SAMHD1* gene was disrupted by CRISPR/Cas9 (THP1-KO). The non-*SAMHD1* expressing AML cell line HEL and the non-*SAMHD1* expressing ALL cell line JURKAT were transduced with wild-type *SAMHD1* (SAMHD1_WT) or the triphosphohydrolase-defective *SAMHD1* mutant D311A (SAMHD1_D311A). Dose-response curves, drug concentrations that reduce cell viability by 50% (IC50s), and Western blots confirming SAMHD1 protein levels are provided. Each symbol represents the mean ± SD of three independent experiments, each performed in three technical replicates.

## Discussion

Similar chemotherapeutic agents are used to treat T-ALL and B-ALL. However, nelarabine is specifically used for relapsed T-ALL[3,7–12]. Although it had been known for decades that nelarabine is more active in T-ALL than in B-ALL cells[6,46], the underlying mechanisms had remained elusive.

Here, we used an approach combining data derived from large pharmacogenomics screens (CTRP, CCLE, GDSC), an RCCL-derived ALL cell line panel, and patient data and found that cellular SAMHD1 levels critically determined ALL cell sensitivity to nelarabine and AraG. Nelarabine is metabolised into AraG, which is then triphosphorylated by cellular kinases

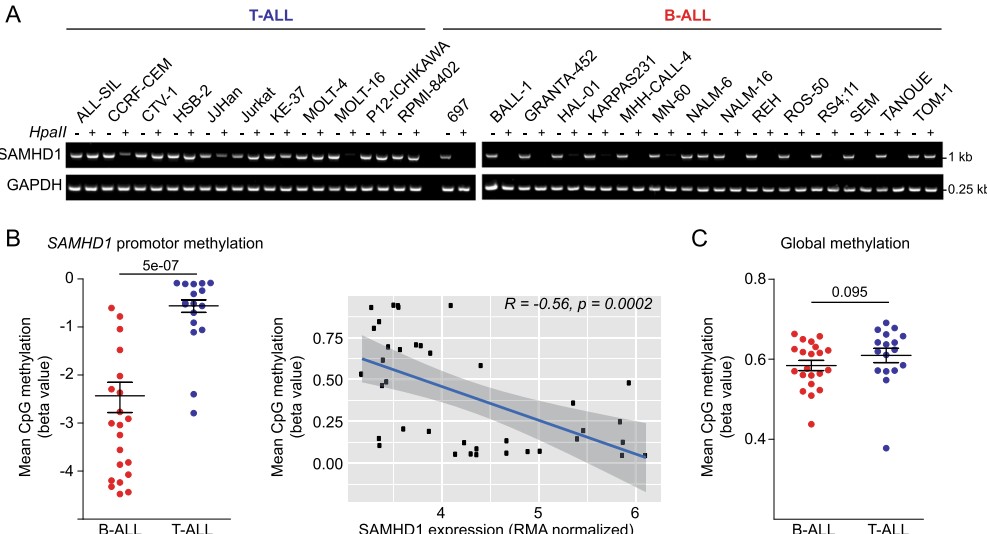

**Fig. 6 SAMHD1 promoter methylation in ALL cell lines. a** Analysis of *SAMHD1* promoter methylation in the RCCL cell line panel through amplification of a single PCR product (993-bp) corresponding to the promoter sequence after HpaII digestion. A 0.25-kb fragment of the GAPDH gene lacking HpaII sites was PCR-amplified using the same template DNA served as loading control. **b** GDSC data indicating *SAMHD1* promoter methylation in B-ALL and T-ALL cell lines and correlation of *SAMHD1* promoter methylation and *SAMHD1* expression across all ALL cell lines. **c** GDSC data indicating the level of global methylation in B-ALL and T-ALL cell lines.

into the active form[40]. SAMHD1 is a deoxynucleotide triphosphate (dNTP) hydrolase that cleaves and inactivates triphosphorylated nucleoside analogues including triphosphorylated AraG[23–25,45]. Moreover, T-ALL cells were characterised by substantially lower SAMHD1 levels than B-ALL cells. Previous studies had demonstrated an association between AraG efficacy and AraG triphosphate levels in leukaemia cells, but the mechanism determining AraG triphosphate levels had remained unknown[6,19,46,47]. Hence, SAMHD1 is the missing link explaining the discrepancy in nelarabine sensitivity between T-ALL and B-ALL.

Notably, SAMHD1 has also been shown to promote DNA damage repair including damage induced by the topoisomerase inhibitors camptothecin and etoposide[48]. Thus, SAMHD1-mediated repair of nelarabine/ AraG-induced DNA damage may potentially also contribute to the increased nelarabine/ AraG resistance associated with high SAMHD1 levels in ALL cells. However, *SAMHD1* expression was not associated with generally increased resistance to DNA damaging agents in the CTRP (Supplementary Data 4). The AUC of the PARP inhibitor veliparib was correlated with *SAMHD1* expression, but the AUC of the PARP inhibitor olaparib was not. The AUCs of etoposide and other prominent DNA damaging agents such as the alkylating agents temozolomide, ifosfamide, and dacarbazine and the nucleoside analogue 5-fluorouracil also were also not correlated with *SAMHD1* expression, and the AUCs of the alkylating agents cyclophosphamide and chlorambucil, the nucleoside analogue gemcitabine, the DNA cross-linker mitomycin C, and the topoisomerase inhibitor doxorubicin displayed a significant inverse correlation to *SAMHD1* expression (Supplementary Data 4). These data do not suggest that SAMHD1 interferes with the effects of anti-cancer drugs predominantly via promotion of DNA damage repair.

Data derived from the RCCL ALL cell line panel as well as from the GDSC indicate that the differences in *SAMHD1* expression observed between T-ALL and B-ALL cells are at least in part the consequence of higher *SAMHD1* promoter methylation in T-ALL than in B-ALL cells. Thus, *SAMHD1* expression levels and *SAMHD1* promoter methylation are potential

biomarkers of nelarabine sensitivity that deserve further clinical investigation. Based on our current data, patients suffering from ALL characterised by high *SAMHD1* expression are unlikely to benefit from therapy using nelarabine and may be better treated with ribose-based thiopurines that are no SAMHD1 substrates, such as 6-thioguanine or 6-mercaptopurine[49].

SAMHD1 depletion sensitised ALL cells to AraG, indicating that SAMHD1 may also serve as a therapeutic target to improve nelarabine therapies in ALL patients. Notably, both T-ALL and B-ALL patients may benefit from SAMHD1 inhibition in combination with nelarabine therapy. Interestingly, the effect of SAMHD1 on the activity of nucleoside analogues varied substantially between different forms of leukaemia. SAMHD1 was previously shown to critically determine the activity of the nucleoside analogue cytarabine in AML[23,24,30]. Compared to the pronounced effects of SAMHD1 on nelarabine/ AraG activity in ALL, however, SAMHD1 exerted only minor effects on the activity of cytarabine in this leukaemia type. Interestingly, the situation was reversed in AML cells, where SAMHD1 critically affected cytarabine activity but had much lower impact on AraG. These findings are important, because they illustrate that, despite a general trend in the biomedical community towards tumour-agnostic approaches, which consider cancer-specific alterations independently of the cancer type[50,51], a much more in depth understanding of the molecular make-up of cancer cells will be required, before therapy decisions can be entirely based on molecular markers without taking the cancer entity into consideration.

In conclusion, our data indicate that cellular SAMHD1 levels critically determine ALL cell sensitivity to nelarabine/ AraG and that T-ALL cells display lower SAMHD1 levels than B-ALL. This provides a solution to a decades old conundrum providing a mechanistic explanation for the higher nelarabine sensitivity of T-ALL cells compared to B-ALL cells. Hence, SAMHD1 has potential as a biomarker for the more accurate identification of ALL patients, who are likely to benefit from nelarabine therapy. Moreover, SAMHD1 is a therapeutic target for the design of improved nelarabine-based treatment strategies for ALL patients.

## Methods

**Analysis of data from pharmacogenomics screens**. ALL cell line drug sensitivity data and RMA-normalised gene expression values were obtained from the CCLE (2015 release, https://portals.broadinstitute.org/ccle), which contains data from 34 ALL cell lines (18 B-ALL and 16 T-ALL)[16], GDSC (2016 release, https://www.cancerrxgene.org/; 21 B-ALL/ 17 T-ALL cell lines)[17], and CTRP (version 2, 2015 release, https://ocg.cancer.gov/programs/ctd2/data-portal; 11 B-ALL/ 13 T-ALL cell lines) [15].

Gene expression was compared using the Mann-Whitney U (Wilcoxon) test for independent groups. Multiple test correction of p-values was performed using the Benjamini–Hochberg (BH) procedure[52], with a false discovery rate (FDR) of 0.05 (BH = (rank/$n$) × FDR, where $n$ = the total number of genes compared). Gene expression levels between B-ALL and T-ALL cell lines were visualised using the ggboxplot function in R. Heatmaps showing gene expression levels were generated using the ggplot2 package in R.

In all, 36 of the ALL cell lines (19 B-ALL, 17 T-ALL) in the GDSC and 22 ALL cell lines (10 B-ALL, 12 T-ALL) in the CTRP were treated with cytarabine. 23 ALL cell lines (11 B-ALL, 12 T-ALL) in the CTRP were treated with nelarabine. Scatter plots and their associated Pearson correlations for each drug AUCs against gene expression were calculated using the ggplot2 package in R.

Expression of 18,542 genes was correlated with the nelarabine AUC in ALL cell lines and SAMHD1 expression was correlated with the AUC values of 441 drugs tested in ALL cell lines using the CTRPv2 dataset. Pearson correlation coefficients were calculated using the cor.test function in R. P-values for each correlation were ranked and multiple test correction was performed (Benjamini–Hochberg procedure, FDR = 0.05).

Pathway analysis was performed using the PANTHER (version 14.1) Overrepresentation Test [Mi et al.[20]] based on genes significantly differentially expressed in B-ALL and T-ALL cell lines after Benjamini–Hochberg p-value correction (FDR = 0.05). Fisher's exact test was applied to calculate over- vs. underrepresentation of classes. Heatmaps were prepared using the ggplot2 package in R.

Beta values for CpG sites in the SAMHD1 promoter derived from the GDSC (Gene Expression Omnibus ID GSE68379) were correlated with SAMHD1 expression in ALL cell lines.

**Analysis of patient data**. *SAMHD1* gene expression was analyzed in publicly available Microarray data of 306 primary adult B- and T-ALL patients (Gene Expression Omnibus ID GSE66006)[28]. The median percentage of leukemic cells in the samples was 90%.

**Drugs**. Cytarabine was purchased from Tocris Biosciences (via Bio-Techne GmbH, Wiesbaden, Germany), AraG from Jena Bioscience (Jena, Germany).

**Cell lines**. The human ALL cell lines 697, ALL-SIL, BALL-1, CTV-1, GRANTA-452, HAL-01, HSB-2, JURKAT, KE-37, MHH-CALL-4, MN-60, MOLT-4, MOLT-16, NALM-6, NALM-16, P12-ICHIKAWA, REH, ROS-50, RPMI-8402, RS4;11, SEM, TANOUE, and TOM-1 and the AML cell lines THP-1 and HEL were obtained from DSMZ (Deutsche Sammlung von Mikroorganismen und Zellkulturen GmbH, Braunschweig, Germany). The ALL cell lines CCRF-CEM and JJHan were received from ATCC (Manassas, VA, US) and the ALL cell line KARPAS231 from Cambridge Enterprise Ltd. (Cambridge, UK).

THP-1 cells deficient in SAMHD1 (THP-1 KO) and control cells (THP-1 Ctr.) were generated using CRISPR/Cas9 approach as previously described[30,45,53]. THP-1 cells were plated at a density of $2 \times 10^5$ cells per ml. After 24 h, $2.5 \times 10^6$ cells were resuspended in 250 µl Opti-MEM, mixed with 5 µg CRISPR/Cas plasmid DNA, and electroporated in a 4-mm cuvette using an exponential pulse at 250 V and 950 mF utilizing a Gene Pulser electroporation device (Bio-Rad Laboratories). We used a plasmid encoding a CMV-mCherry-Cas9 expression cassette and a human SAMHD1 gene specific gRNA driven by the U6 promoter. An early coding exon of the SAMHD1 gene was targeted using the following gRNA construct: 5′-CGGAAGGGGTGTTTGAGGGG-3′. Cells were allowed to recover for 2 days in six-well plates filled with 4 ml medium per well before being FACS sorted for mCherry-expression on a BD FACS Aria III (BD Biosciences). For subsequent limiting dilution cloning, cells were plated at a density of 5, 10, or 20 cells per well of nine round-bottom 96-well plates and grown for 2 weeks. Plates were scanned for absorption at 600 nm and growing clones were identified using custom software and picked and duplicated by a Biomek FXp (Beckman Coulter) liquid handling system.

The HEL and JURKAT SAMHD1-WT and SAMHD1-D311A cell lines were generated by co-transfection of the packaging vector pPAX2 (Addgene), either pHR-SAMHD1-WT or pHR-SAMHD1-D311A and a plasmid encoding VSV-G, as previously described[45]. All cell lines were routinely tested for Mycoplasma, using the MycoAlert PLUS assay kit from Lonza, and were authenticated by short tandem repeat profiling, as described elsewhere.

All cell lines were cultured in IMDM (Biochrom) supplemented with 10% FBS (Sigma-Aldrich), 4 mM L-glutamine (Sigma-Aldrich), 100 IU per ml penicillin (Sigma-Aldrich), and 100 µg per ml streptomycin (Sigma-Aldrich) at 37 °C in a humidified 5% $CO_2$ incubator.

**Viability assay**. Cell viability was determined by 3-(4,5-dimethylthiazol-2-yl)-2,5-diphenyltetrazolium bromide (MTT) assay modified after Mosman[54], as previously described[55]. Cells suspended in 100 µL cell culture medium were plated per well in 96-well plates and incubated in the presence of various drug concentrations for 96 h. Then, 25 µL of MTT solution (2 mg/mL (w/v) in PBS) were added per well, and the plates were incubated at 37 °C for an additional 4 h. After this, the cells were lysed using 200 µL of a buffer containing 20% (w/v) sodium dodecylsulfate in 50% (v/v) N,N-dimethylformamide with the pH adjusted to 4.7 at 37 °C for 4 h. Absorbance was determined at 570 nm for each well using a 96-well multiscanner. After subtracting of the background absorption, the results are expressed as percentage viability relative to control cultures which received no drug. Drug concentrations that inhibited cell viability by 50% (IC50) were determined using CalcuSyn (Biosoft, Cambridge, UK).

**Western blotting**. Western blotting was performed as previously described [Schneider et al.[30]]. Cells were lysed in Triton X-100 sample buffer and proteins separated by sodium dodecyl sulfate-polyacrylamide gel electrophoresis. Proteins were blotted on a nitrocellulose membrane (Thermo Scientific). The following primary antibodies were used at the indicated dilutions: SAMHD1 (Proteintech, 12586-1-AP, 1:1,000), β-actin (BioVision, 3598R-100, 1:5,000), pSAMHD1 (Cell Signaling, 89930S, 1:1,000), and GAPDH (Trevigen, 2275-PC-10C, 1:5,000). Visualisation and quantification were performed using IRDye-labeled secondary antibodies (LI-COR Biotechnology, IRDye®800CW Goat anti-Rabbit, 926-32211, 1:40,000) according to the manufacturer's instructions. Band volume analysis was conducted by Odyssey LICOR. Uncropped blots are presented in Supplementary Fig. 10. SAMHD1 quantification was performed using a protein extract of the AML cell line THP-1 as internal control (Supplementary Fig. 10, Supplementary Data 5).

**mRNA analysis**. RNA extraction and TaqMan-based mRNA quantification of SAMHD1 (assay no. Hs00210019_m1) and RNaseP (TaqMan® RNaseP Assay (A30065)) as endogenous reference control were performed according to the manufactures protocol (Applied Biosystems). Total RNA was extracted using the RNeasy Kit from Qiagen and stored at −80 °C until use. Relative quantitative PCR analyses were performed on the QuantStudio 7 Flex Real-Time PCR System (Applied Biosystems). SAMHD1 mRNA expression levels were quantified by using the ΔΔCt method with RNaseP mRNA as an endogenous reference control. All samples were run in triplicate. Data analysis was conducted using the QuantStudio System Software (Applied Biosystems).

***SAMHD1* promoter methylation**. *SAMHD1* promoter methylation was determined as previously described[45]. *SAMHD1* promoter contains five HpaII sites surrounding the transcription start site[44]. Methylation of the HpaII sites in the SAMHD1 promoter would prevent digestion by the HpaII, and the intact sequence would serve as a template for PCR amplification using SAMHD1 promoter-specific primers that flank the HpaII sites. To measure methylation of the SAMHD1 promoter genomic DNA was treated with the methylation-sensitive HpaII endonuclease or left untreated as described previously with some modifications[44]. PM3. fwd: TTCCGCCTCATTCGTCCTTG and PM3.rev: GGTTCTCGGGCTGTCATCG were used as SAMHD1 promoter-specific primers. A single PCR product (993-bp) corresponding to the SAMHD1 promoter sequence was obtained from untreated genomic DNA and treated DNA from cells with methylated but not from cells with unmethylated SAMHD1 promoter. To serve as input control, a 0.25-kb fragment of the GAPDH gene lacking HpaII sites was PCR-amplified using the same template DNA[44]. Uncropped agarose gels are shown in Supplementary Fig. 10.

**SAMHD1 depletion using Vpx virus-like particles**. Cells were spinoculated with VSV-G pseudotyped virus-like particles carrying either Vpx or Vpr from SIV-mac251, produced by co-transfection of 293T cells with pSIV3 + gag pol expression plasmids and a plasmid encoding VSV-G as previously described [30,44].

**Statistics and reproducibility**. Statistical data analyses were performed in GraphPad Prism version 7. Population means were compared using unpaired two-tailed Student's t-tests. Data are presented as means ± standard deviation (S.D.). Specific information on the number and nature of replicates is provided in the figure legends. Correlation analyses were performed using linear regression in GraphPad Prism resulting in $r^2$ as a measure for goodness-of-fit and the P value, which is calculated from an F test, indicating whether the slope is significantly different from zero.

**Reporting summary**. Further information on research design is available in the Nature Research Reporting Summary linked to this article.

## Data availability

All source data underlying the graphs, which are not based on publicly available data from CCLE, CTRP, GDSC and patient gene expression data (Gene Expression Omnibus

ID GSE66006), are presented in Supplementary Data 5. All other data are present in the paper or available from the corresponding authors upon reasonable request.

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

## Acknowledgements
We thank Sebastian Grothe, Anja Hüttinger, Lena Stegmann, and Eva Wagner for technical assistance. The work was supported by the Frankfurter Stiftung für krebskranke Kinder and the Hilfe für krebskranke Kinder Frankfurt e.V. Tobias Herold is supported by the Physician Scientists Grant (G-509200-004) from the Helmholtz Zentrum München.

## Author contributions

T.R., K.M., C.S., F.R., and J.C. have performed experiments. T.R., K.M., T.H., C.S., T.O., F.R., A.F., T.R.F., M.N.W., O.T.K., M.M. and J.C. have analysed data. M.M., M.N.W., and J.C. have supervised research. M.M. and J.C. have conducted the project. M.M. has written the initial draft. All authors have contributed to drafting the final paper. T.R., K.M., T.H., C.S., T.O., F.R., A.F., T.R.F., M.N.W., O.T.K., M.M. and J.C. have proofread the paper and agree with its content.

## Competing interests

The Johann Wolfgang Goethe University has filed a patent application, on which T.O., C.S., O.T.K. and J.C. are listed as inventors. All other authors declare no competing interests.
