## [Peer Review File · Communications Biology]

Reviewers' comments:

Reviewer #1 (Remarks to the Author):

The manuscript is well written concise and provide new insight in the mechanism of response to nelarabine especially in T-ALL. Given that only 30% of relapsed/refractory T-ALL respond to nelarabine, the identification of potential biomarkers of response will be of great interest in a clinical setting. Considering that several trials are considering nelarabine for induction chemotherapy this study has the potential to provide relevant information.

The reviewer thinks integrating multiple computational approaches is clever and deserves probably a better representation in Figure 1 (a schema of how data has been integrated).

The major limitations are the extensive use of cell lines or data derived from publicly available data set. The authors generated data on primary cells from Microarray previously published ID 66006

- One idea is to probe for SAMHD1 in T-ALL bone marrow biopsies before and after Nelarabine treatment and correlate results with a clinical response. In fact, the reviewer believes it will be of interested great interest test for SAMHD1 in R/R T-ALL that failed nelarabine rescue compared to the one achieved complete remission.

- A second effort is to test for protein expression of SAMHD1 in freshly isolated clinical leukemia samples.

The quality of Figure 2 and 6B is not as good as the rest of panels. It looks blurry. Please revise accordingly.

Reviewer #2 (Remarks to the Author):

In this interesting manuscript, Rothenburger and colleagues described SAMHD1 as the key mediator of the therapeutic effects of Nelarabine in T- vs. B-ALL. Taking advantage of published data from CCLE, CTRP and GDSC, authors first identify the levels of SAMHD1 as the gene with most significant correlation between expression and therapeutic effect of Nelarabine. Consistently, B-ALL cell lines showed higher levels of SAMHD1 than T-ALL cell lines, similar to what could be observed in patients. Authors then showed that SAMHD1 knockdown sensitizes ALL cells to Nelarabine, and over expression of an SAMHD1 mutant version also resulted in higher sensitivity to Nelarabine as compared to SAMHD1 WT. Finally, authors show that T-ALL cell lines show general hypermethylation of the SAMHD1 promoter, whereas most B-ALL cell lines don't show this effect. Consistently, the only T-ALL cell line showing some SAMHD1 expression didn't show methylation of its promoter, and the 2 B-ALL cell lines that showed promoter methylation are amongst the B-ALL cell lines with lower SAMHD1 levels. I think this study merits publication. Still, I have some minor comments that authors should address:

1. In Fig 1B, authors show that T-ALL patients show lower SAMHD1 mRNA levels as compared to B-ALL patients. Can they also check the relevant genes investigated in cell lines in Fig. 1C but in this same cohort of patient samples, to check whether that lack of correlation still holds?

2. Since AML and T-ALL cells respond differently to Nelarabine, can authors show how SAMHD1 levels compare between AML and T-ALL cell lines and/or patients? As of now authors only discuss that differences in the tissue of origin might explain these results, but it should be relatively easy to check how the total SAMHD1 levels compare between these hematological malignancies, and whether these are consistent with what they observe.

3. In Figure 3A, it looks like 3 independent WBs are shown. Authors should also perform additional WBs in which they compare concomitantly B-ALL and T-ALL samples in the same membrane. If not,

the big differences observed could be due just to different exposure times, etc.

4. In Fig 5B, authors should perform at the same time the dose-response curves in the parental cells, together with cells over expressing mutant and WT SAMHD1. As shown now, authors can't state that SAMHD1 overexpression induces Ara-G/Cytarabine resistance, but rather that the SAMHD1 mutant version sensitizes to these drugs, as compared to SAMHD1 WT. All three cell lines should be shown concomitantly.

Reviewer #3 (Remarks to the Author):

This is a very nice study on the role of SAMHD1 levels in determining response to nelarabine. The authors convincingly demonstrate that T-ALL cases have lower SAMHD1 levels than B-ALL cases and that this correlates with response to nelarabine. They use data from patient samples and cell lines. Overexpression of SAMHD1 in sensitive cell lines clearly makes the cells less sensitive and the enzymatic activity of SAMHD1 is needed.

This study is well performed, clearly described and is clinically very relevant.

I only have one suggestion for improvement:

Figure 5:

It would be easier to understand the effects of SAMHD1 overexpression if the dose response curves would be shown for the parental cells (or empty vector transduced cells ideally): for example for Jurkat cells: show also the dose response curve for the parental cells so that it is easy to compare with SAMHD WT overexpressing cells and with mutant SAMHD1 expressing cells.

Reviewer #1 (Remarks to the Author):

The manuscript is well written concise and provide new insight in the mechanism of response to nelarabine especially in T-ALL. Given that only 30% of relapsed/refractory T-ALL respond to nelarabine, the identification of potential biomarkers of response will be of great interest in a clinical setting. Considering that several trials are considering nelarabine for induction chemotherapy this study has the potential to provide relevant information. The reviewer thinks integrating multiple computational approaches is clever and deserves probably a better representation in Figure 1 (a schema of how data has been integrated).

The major limitations are the extensive use of cell lines or data derived from publicly available data set. The authors generated data on primary cells from Microarray previously published ID 66006

- One idea is to probe for SAMHD1 in T-ALL bone marrow biopsies before and after Nelarabine treatment and correlate results with a clinical response. In fact, the reviewer believes it will be of interested great interest test for SAMHD1 in R/R T-ALL that failed nelarabine rescue compared to the one achieved complete remission.
- A second effort is to test for protein expression of SAMHD1 in freshly isolated clinical leukemia samples.

Authors' response:

Although, we agree that having these data would be great, we do not think that it is feasible to generate meaningful data on this in a revision. With regard to the first suggestion there are probably less than 12 patients a year in countries like Germany or the UK. Due to the small number of patients, the second approach would still leave us, even in the best case, with a very limited number of samples, which would be of limited impact. We have analysed the SAMHD1 mRNA levels in samples of 306 ALL patients and also shown that SAMHD1 mRNA levels correlate with SAMHD1 protein levels. Hence, we do not feel that adding a limited amount of additional samples would increase the impact. Taken together, we think that the suggested experiments describe a larger, probably multi-centric prospective study (if they are meant to generate meaningful data) and not a revision. Thus, they are beyond the scope of this preclinical study.

The quality of Figure 2 and 6B is not as good as the rest of panels. It looks blurry. Please revise accordingly.

Authors' response:

This has been done.

Reviewer #2 (Remarks to the Author):

In this interesting manuscript, Rothenburger and colleagues described SAMHD1 as the key mediator of the therapeutic effects of Nelarabine in T- vs. B-ALL. Taking advantage of published data from CCLE, CTRP and GDSC, authors first identify the levels of SAMHD1 as the gene with most significant correlation between expression and therapeutic effect of Nelarabine. Consistently, B-ALL cell lines showed higher levels of SAMHD1 than T-ALL cell lines, similar to what could be observed in patients. Authors then showed that SAMHD1 knockdown sensitizes ALL cells to Nelarabine, and over expression of an SAMHD1 mutant version also resulted in higher sensitivity to Nelarabine as compared to SAMHD1 WT. Finally, authors show that T-ALL cell lines show general hypermethylation of the SAMHD1 promoter, whereas most B-ALL cell lines don't show this effect. Consistently, the only T-ALL cell line showing some SAMHD1 expression didn't show methylation of its promoter, and the 2 B-ALL cell lines that showed promoter methylation are amongst the B-ALL cell lines with lower SAMHD1 levels. I think this study merits publication. Still, I have some minor comments that authors should address:

1. In Fig 1B, authors show that T-ALL patients show lower SAMHD1 mRNA levels as compared to B-ALL patients. Can they also check the relevant genes investigated in cell lines in Fig. 1C but in this same cohort of patient samples, to check whether that lack of correlation still holds?

Authors' response:

We have done this and have detected additional significant changes in the patient data set than in the cell line data set. In addition to SAMHD1, we also detected relevant significant differences in DGUOK, SLC29A1, NT5C2, and PNP (NT5C displayed significantly higher expression in T-ALL relative to B-ALL, if it were expected to be responsible for increased T-ALL sensitivity, it would have been expected to be the other way around). However, the differences in SAMHD1 expression were by far the most pronounced ones. The respective data was included in Suppl. Figure 4, and the respective paragraph was amended as follows (p. 8, lines 140-152):

"A number of other gene products have been described to be involved in the transport, activation, and metabolism of nucleoside analogues such as nelarabine, including DCK, DGUOK, SLC29A1 (ENT1), SLC29A2 (ENT2), NT5C, NT5C2, PNP, RRM1, RRM2, and SLC22A4 (OCTN1) [Homminga et al., 2011; Drenberg et al., 2017]. While statistically significant differences in the expression of some of the respective genes were noted between B-ALL and T-ALL cell lines in some of the three datasets, none was consistent across all three and none was as robust as in the expression of SAMHD1 (Figure 1C, Suppl. Figure 4). In patient samples, SAMHD1 also displayed the most significant difference in expression levels between B-ALL and T-ALL (Suppl. Figure 4). Moreover, only the expression of SAMHD1 correlated with the nelarabine AUC in the CTRP dataset (Figure 2, Suppl. Figure 5). This shows that SAMHD1 is a critical determinant of nelarabine efficacy in ALL and that low SAMHD1 levels critically contribute to the specific nelarabine sensitivity of T-ALL cells."

2. Since AML and T-ALL cells respond differently to Nelarabine, can authors show how SAMHD1 levels compare between AML and T-ALL cell lines and/or patients? As of now authors only discuss that differences in the tissue of origin might explain these results, but it should be relatively easy to check how the total SAMHD1 levels compare between these hematological malignancies, and whether these are consistent with what they observe.

Authors' response:

This comparison could be done for cell lines using GDSC, CCLE, and CTRP data. It has shown that T-ALL cells display lower SAMHD1 expression than AML cells. The results are presented in the novel Suppl. Figure 6. The text was amended as follows (p. 9, lines 163-166):

"Cellular SAMHD1 levels have previously been shown to critically determine cytarabine efficacy in acute myeloid leukaemia (AML) cells [Hollenbaugh et al., 2017; Schneider et al., 2017; Knecht et al., 2018] and *SAMHD1* expression levels are lower in T-ALL than in AML cells (Suppl. Figure 6)."

3. In Figure 3A, it looks like 3 independent WBs are shown. Authors should also perform additional WBs in which they compare concomitantly B-ALL and T-ALL samples in the same membrane. If not, the big differences observed could be due just to different exposure times, etc.

Authors' response:

We have used a lysate of the AML cell line THP-1 as internal control for SAMHD1 protein quantification. We should have explained this better in our Methods section and apologise for this. It is now mentioned (“SAMHD1 quantification was performed using a protein extract of the AML cell line THP-1 as internal control (Suppl. Figure 11, Suppl. Table 7).”, p. 28, lines 517-519) and shown in the new Suppl. Figure 11 and the new Suppl. Table 7.

To confirm these results further, we directly compared 23 of the 26 cell lines on one gel, which was blotted on one membrane. The results are confirmed the results from Figure 3A and are presented as Suppl. Figure 8 (below) in the revised version of the manuscript.

Suppl. Figure 8

**For comparison:
Figure 3A**

Suppl. Figure 8. SAMHD1 protein levels in the RCCL panel of B-ALL and T-ALL cell lines. Representative Western blots indicating protein levels of total SAMHD1 and GAPDH in 23 cell lines of the RCCL panel, which were run on the same gel and blotted on the same membrane to confirm the representativeness of the blots provided in Figure 3A. Figure 3A is provided for comparison.

4. In Fig 5B, authors should perform at the same time the dose-response curves in the parental cells, together with cells over expressing mutant and WT SAMHD1. As shown now, authors can't state that SAMHD1 overexpression induces Ara-G/Cytarabine resistance, but rather that the SAMHD1 mutant version sensitizes to these drugs, as compared to SAMHD1 WT. All three cell lines should be shown concomitantly.

Authors' response:

This has been done. The updated Figure 5 is presented below.

Figure 5

Reviewer #3 (Remarks to the Author):

This is a very nice study on the role of SAMHD1 levels in determining response to nelarabine. The authors convincingly demonstrate that T-ALL cases have lower SAMHD1 levels than B-ALL cases and that this correlates with response to nelarabine. They use data from patient samples and cell lines. Overexpression of SAMHD1 in sensitive cell lines clearly makes the cells less sensitive and the enzymatic activity of SAMHD1 is needed. This study is well performed, clearly described and is clinically very relevant.

I only have one suggestion for improvement:

Figure 5:

It would be easier to understand the effects of SAMHD1 overexpression if the dose response curves would be shown for the parental cells (or empty vector transduced cells ideally): for example for Jurkat cells: show also the dose response curve for the parental cells so that it is easy to compare with SAMHD WT overexpressing cells and with mutant SAMHD1 expressing cells.

Authors' response:

This was done. The updated Figure 5 is presented below.

Figure 5

REVIEWERS' COMMENTS:

Reviewer #2 (Remarks to the Author):

Authors have been responsive to my comments and the paper has overall improved. I recommend it for publication, and I congratulate the authors for their nice and clinically relevant story.

Reviewer #3 (Remarks to the Author):

Very nice study, the authors have now provided a revised manuscript that is further improved. I have no new comments.